# Microendoscope-Assisted Versus Open Posterior Lumbar Interbody Fusion for Lumbar Degenerative Disease: A Multicenter Retrospective Cohort Study

**DOI:** 10.3390/medicina57020150

**Published:** 2021-02-08

**Authors:** Masayoshi Fukushima, Nozomu Ohtomo, Michita Noma, Yudai Kumanomido, Hiroyuki Nakarai, Keiichiro Tozawa, Yuichi Yoshida, Ryuji Sakamoto, Junya Miyahara, Masato Anno, Naohiro Kawamura, Akiro Higashikawa, Yujiro Takeshita, Hirohiko Inanami, Sakae Tanaka, Yasushi Oshima

**Affiliations:** 1Spine Center, Toranomon Hospital, Tokyo 105-8470, Japan; masamasayoshi21@gmail.com (M.F.); m.noma@me.com (M.N.); kumanomidoh2@yahoo.co.jp (Y.K.); masato.anno@gmail.com (M.A.); 2University of Tokyo Spine Group (UTSG), Tokyo 113-8655, Japan; n.ohtomo1002@gmail.com (N.O.); NAKARAIH-ORT@h.u-tokyo.ac.jp (H.N.); le_parc_de_joie@yahoo.co.jp (K.T.); yoshi.med.red@gmail.com (Y.Y.); under-age-songs@hotmail.co.jp (R.S.); junya.miyahara@gmail.com (J.M.); naohirokawamura@yahoo.co.jp (N.K.); ahigashi-tky@umin.ac.jp (A.H.); bamboo4ta@gmail.com (Y.T.); inanamihiro@gmail.com (H.I.); 3Department of Orthopaedic Surgery, Faculty of Medicine, The University of Tokyo, Tokyo 113-8655, Japan; TANAKAS-ORT@h.u-tokyo.ac.jp; 4Department of Spine and Orthopedic Surgery, Japanese Red Cross Medical Center, Tokyo 150-8935, Japan; 5Department of Orthopaedic Surgery, Japan Organization of Occupational Health and Safety Kanto Rosai Hospital, Kanagawa 211-8510, Japan; 6Department of Orthopaedic Surgery, Japan Organization of Occupational Health and Safety, Yokohama Rosai Hospital, Kanagawa 222-0036, Japan; 7Inanami Spine and Joint Hospital, Tokyo 140-0002, Japan

**Keywords:** microendoscope, posterior lumbar interbody fusion, multicenter study

## Abstract

*Background and objectives*: Minimally invasive surgery has become popular for posterior lumbar interbody fusion (PLIF). Microendoscope-assisted PLIF (ME-PLIF) utilizes a microendoscope within a tubular retractor for PLIF procedures; however, there are no published reports that compare Microendoscope-assisted to open PLIF. Here we compare the surgical and clinical outcomes of ME-PLIF with those of open PLIF. *Materials and Methods*: A total of 155 consecutive patients who underwent single-level PLIF were registered prospectively. Of the 149 patients with a complete set of preoperative data, 72 patients underwent ME-PLIF (ME-group), and 77 underwent open PLIF (open-group). Clinical and radiographic findings collected one year after surgery were compared. *Results*: Of the 149 patients, 57 patients in ME-group and 58 patients in the open-group were available. The ME-PLIF procedure required a significantly shorter operating time and involved less intraoperative blood loss. Three patients in both groups reported dural tears as intraoperative complications. Three patients in ME-group experienced postoperative complications, compared to two patients in the open-group. The fusion rate in ME-group at one year was lower than that in the open group (*p* = 0.06). The proportion of patients who were satisfied was significantly higher in the ME-group (*p* = 0.02). *Conclusions:* ME-PLIF was associated with equivalent post-surgical outcomes and significantly higher rates of patient satisfaction than the traditional open PLIF procedure. However, the fusion rate after ME-PLIF tended to be lower than that after the traditional open method.

## 1. Introduction

Posterior lumbar interbody fusion (PLIF) is a commonly used surgical intervention to treat degenerative spinal disorders. PLIF has become very popular among surgeons because it provides a relatively large surface area for fusion with a graft inserted between adjacent vertebral bodies [1,2,3]. Traditional open techniques used for PLIF could lead to extensive tissue damage [4]; as such, new adjustments to the procedure have resulted in minimally invasive PLIF (MI-PLIF) [5], in which spinal decompression and cage insertion can be performed with a smaller skin incision, followed by percutaneous pedicle screw insertion. Several studies have documented the outcomes of the MI-PLIF, which permits the posterior musculature to be maintained, in terms of minimized blood loss, decreased back pain, and shorter hospital stay [5]. Nonetheless, there remain concerns regarding impaired interbody bone fusion due to the limited area of endplate curettage and bone graft.

Microendoscope-assisted PLIF (ME-PLIF), one of the MI-PLIF techniques, is performed using a microendoscope for the decompression and cage insertion. Originally, a microendoscopic discectomy was developed as a treatment for herniation of the lumbar disks by Foley and Smith; this procedure which utilizes a 16 mm or 18 mm tubular retractor with an internal microscope [6]; favorable clinical outcomes using this device have been reported [7,8]. This procedure has been applied to decompression surgery for degenerative cervical or lumbar spinal disorders [9,10,11] and also for PLIF [12,13]. However, there are no published reports compared clinical outcomes from ME-PLIF with those of conventional open PLIF. The purpose of this study was to compare clinical and radiographic outcomes of ME-PLIF and open PLIF for single-level degenerative lumbar diseases.

## 2. Material and Methods

### 2.1. Study Design

This is a multicenter retrospective cohort study that enrolled patients prospectively registered who underwent single-level PLIF for lumbar degenerative disorders at one of six high volume spine centers between April 2017 and June 2018. Patients with a history of previous lumbar surgery, tumor, infection, rheumatoid arthritis, or scoliosis (Cobb angle ≥10° on neutral radiographs) were excluded from this study. This study was approved by the institutional review boards of each of the six hospitals. Additionally, written informed consent was obtained from all participants.

A total of 155 consecutive patients who met the aforementioned inclusion criteria were registered for the study. Six of these patients lacked complete preoperative data; therefore, 149 patients were eligible for analysis. Of these patients, 72 underwent ME-PLIF (ME group), and 77 underwent open PLIF (open group); the specific procedure was determined at each hospital. Two facilities performed ME-PLIF on all patients with the single-level disease, while three facilities performed open PLIF on all patients; at a sixth facility, both procedures were performed. ME-PLIF procedures were performed in accordance with previously reported guidelines [12]. First, a 20 mm incision was made into the skin 15 mm symptomatic outside from the midline on the level of the intervertebral disc. Subsequently, using fluoroscopic guidance, a METRx (18 mm, Medtronic Sofamor Danek., Dublin, Ireland) or ESD II tubular retractor (20 mm, Japan Medical Dynamic Marketing, Inc., Tokyo, Japan) was placed at a site overlying the disk space. All procedures leading to interbody fusion, including decompression, removal of an intervertebral disc, grafting of autologous bone and cage insertion, were performed within the microendoscopic field connected to the tubular retractor. Either a polyetheretherketone (PEEK) or titanium interbody cage filled with autologous bone was inserted. After the microendoscopic procedure, pedicle screws were inserted percutaneously under fluoroscopic guidance. We indicated the perioperative and postoperative radiographs, intraoperative microendoscopic surgical field and postoperative wound in patients who underwent ME-PLIF (Figure 1). The traditional open PLIF was performed via a posterior midline incision. After unilateral or bilateral decompression of the spinal canal, grafting of autologous bone and cage insertion was performed at one or both sides. Pedicle screws were inserted from the same surgical field.

All patients were followed for at least one year. Postoperative complications, including additional surgeries, were recorded at each patient visit. Clinical outcomes included a numerical rating scale for low back pain and leg pain, the Oswestry disability index, EuroQol 5 Dimension (EQ-5D) and patients’ overall satisfaction with the treatment. A 7-point Likert scale was used to determine patient satisfaction, with possible answers including “very dissatisfied”, “dissatisfied”, “somewhat dissatisfied”, “unsure”, “somewhat satisfied”, “satisfied”, and “very satisfied.” We defined the patients who answered “somewhat satisfied”, “satisfied”, or “very satisfied” as patients who were overall satisfied with the surgery.

Serial radiographs (neutral and dynamic standing films) and computed tomography (CT) scans were performed one year after surgery. A successful fusion was defined by fulfilling the criteria of both dynamic radiographs and CT scans, including translation of less than or equal to 3 mm and angular motion of less than or equal to 5° on flexion/extension on lateral lumbar radiographs and the presence of bridging trabecular bone within the disk space in the absence of loosening of the pedicle screws (i.e., a ‘‘halo” around the screws) on CT [14,15]. The presence of fusion was judged by two independent physicians (M.F. and N.O.).

### 2.2. Statistical Analysis

Statistical evaluation was performed using JMP software version 12.0 (SAS Institute Inc., Cary, NC, USA). A Mann–Whitney U test was used for nonparametric data, and the chi-squared or Fisher’s exact test was used for categorical variables. A *p* value less than 0.05 was considered statistically significant.

## 3. Results

One hundred and fifteen (77%) of the patients were available for follow-up at one year postoperatively, including 57 (79%) of the 72 patients who underwent ME-PLIF and 58 (75%) of the 77 patients who underwent open PLIF. The demographic data for each group is shown in Table 1. There were no significant differences in the baseline characteristics between the two groups.

The findings shown in Table 2 include a comparison of perioperative data between the ME and open groups. The operating time required for the ME-PLIF procedure was significantly shorter and involved a significantly smaller amount of blood loss than typically experienced in the open PLIF procedure. However, there were no significant differences in terms of intraoperative complications, including three patients with dural tears in each group. During the postoperative follow-up period, three patients with ME-PLIF required additional surgery because of loosening of the cage, loosening of the screws, or infection; one patient who underwent open PLIF suffered from postoperative hematoma and also required additional surgery. One patient who underwent open PLIF developed an early vertebral body fracture after surgery, which healed with conservative treatment (Table 2). With respect to radiographic analysis, 35 of the 57 patients in the ME group (61.4%) developed interbody fusion at one year compared to 45 out of the 58 patients (77.6%) in the open group (*p* = 0.06).

The mean values of the patient-reported outcomes are shown in Table 3. Patients in both groups reported postoperative improvement with no significant differences between those in the ME and those in the open groups. Patient satisfaction was 89.4% (51/57) in ME Group, although only 70.1% (41/58) in open group (*p* = 0.02).

## 4. Discussion

Our aim was to clarify clinical and radiographic outcomes of ME-PLIF and conventional open PLIF. Our study has revealed two main findings. First, the ME-PLIF procedure required a shorter operating time and was associated with significantly less blood loss. Second, although the extent of fusion tended to be lower in the patients undergoing ME-PLIF at one year after surgery, the clinical outcomes were equivalent, and the patient satisfaction rate was higher.

There are many procedures currently in use that can promote lumbar interbody fusion; many recent clinical studies have evaluated the minimally invasive approaches [16]. Recently, minimally invasive posterior lumbar interbody fusion (MI-PLIF) performed with microscope assistance has become popular for lumbar interbody fusion. Several meta-analyses revealed that the short-term outcomes of MI-PLIF, including estimated blood loss, duration of hospital stay, need for narcotics for pain relief, and time until mobilization, were all improved in comparison with the open PLIF, most likely due to the less-invasive nature of the procedure [17,18,19]. However, the same meta-analyses reported that the long-term outcomes, including fusion and complication rates for both MI-PLIF and open PLIF, were similar to one another. In this study, we confirmed that the ME-PLIF procedure required shorter operating times than open PLIF. Interestingly, previous studies and meta-analyses have reported that the surgical times for the two procedures were indistinguishable [18,19]. We speculate that the microendoscopic technique featured here provides an improved visual field compared to those used in other minimally invasive techniques because, in this case, the “eyes” are placed directly within the body; the improved visual fields may lead to shorter operating times. In addition, the more inexperienced surgeons tend to select open procedures; this may be another reason why open PLIF was significantly more time-consuming. Nevertheless, it is reasonable to assume that any minimally invasive procedure would take less time to complete once significant skill has been acquired, if only because they do not require extensive and sometimes difficult exploration of the posterior supporting structures.

A few reports have noted that microendoscopic spinal surgery requires extensive training and experience because of the technical difficulties involved [20,21]. In this study, there were no differences with respect to intraoperative and postoperative complications. We speculate that ME-PLIF was performed by surgeons who had mastered microendoscopic techniques for decompression surgery for disc hernia or lumbar spinal stenosis. However, recent meta-analyses reported that MI-PLIF tended to be associated with a comparatively high revision rate; these findings suggest that there may be a steep learning curve for the skills and abilities required [17]. Therefore, surgeons who are familiar with the spinal microendoscopic technique after having mastered open procedures should start taking the time to become skilled at ME-PLIF so that these procedures can be completed safely without serious complications.

We also found that the fusion rate in patients undergoing ME-PLIF was lower one year after surgery than those undergoing open PLIF, although this difference did not reach statistical significance. Past reports revealed that the rate of bone grafting in association with MI-PLIF was lower than open PLIF because of the limited surgical field [22]. We speculate that the lower fusion rate in this study may be directly related to the lower area of bone grafting due to the unilateral approach and the limited surgical field.-Likewise, for reasons unclear, the fusion rates in patients in both groups in this study were lower than those reported in previous studies [17,18,19]. Most of the previous reports featured postoperative dynamic radiographs rather than CT to evaluate the fusion rate [19,22]; both modalities were used in this study, which may explain the differences observed. Nevertheless, surgeons should keep in mind that the fusion rate after ME-PLIF could be lower than that reported for the traditional open method.

Finally, we found that ME-PLIF was associated with clinical outcomes and implant failure rate after surgery that were indistinguishable from those observed after open PLIF. Several previous reports also showed that there was not always a clear correlation between the fusion rate and clinical outcomes [23,24,25]; one previous study reported that the fusion might be delayed with ultimately satisfactory clinical results [25]. Furthermore, our study revealed that patients who underwent ME-PLIF had a better satisfaction rate at one year after surgery than patients who underwent open PLIF. Past studies reported that minimally invasive PLIF was an effective and safe method with the advantages of reduced operative time and blood loss and limited damage to paravertebral muscles and bone structures [4]. Although postoperative patient-reported outcomes were not significant in this study, perioperative pain or quality of life issues might have had an immediate impact on patient satisfaction with their respective treatments.

There are several limitations to this study. First, due to the nature of a multicenter study with many surgeons, there may be differences in surgical indications and procedures, which may have an unanticipated impact on the results. Second, the follow-up rate was only one year. Longer follow-up of these patients may lead to different results, notably secondary to implant failure or adjacent segmental pathologies that emerge in some patients. Finally, radiological assessments were not performed blindly; as such, this may have introduced inadvertent errors in scoring the results. However, this study has a significant strength in that both dynamic radiography and CT were used to judge bone fusion. Further studies will be necessary to elucidate these issues.

## 5. Conclusions

ME-PLIF was associated with equivalent post-surgical outcomes and significantly higher rates of patient satisfaction than the traditional open PLIF procedure. However, the fusion rate after ME-PLIF tended to be lower than that after the traditional open method.

## Figures and Tables

**Figure 1 medicina-57-00150-f001:**
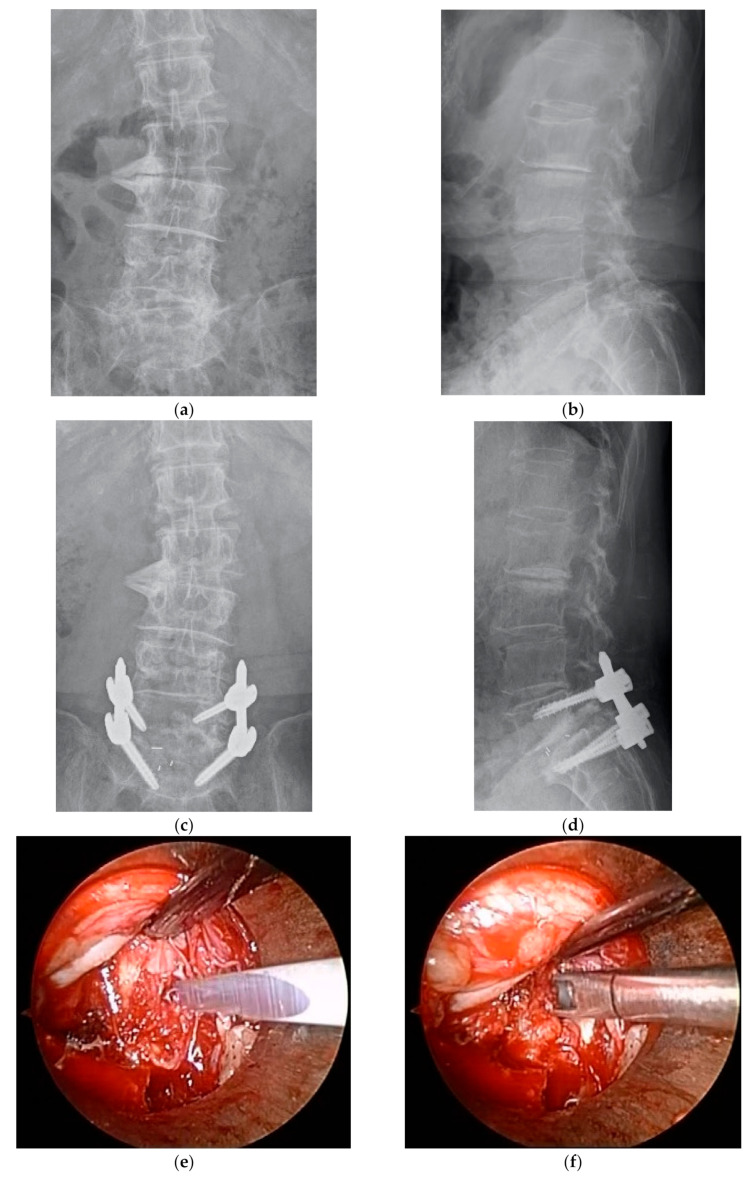
(**a**) Preoperative anterior-posterior (AP) radiograph. (**b**) Preoperative lateral radiograph. (**c**) Postoperative AP radiograph. (**d**) Postoperative lateral radiograph. (**e**) Removal of the intervertebral disc, intraoperative microendoscopic surgical field. (**f**) Curettement of the intervertebral disc, intraoperative microendoscopic surgical field. (**g**) Cage insertion, intraoperative microendoscopic surgical field. (**h**) Postoperative wound.

**Table 1 medicina-57-00150-t001:** Demographic data of patients in microendoscope-assisted (ME) and open groups.

	ME Group	Open Group	*p*
Follow up rate (%)	79% (57/72)	75% (58/77)	0.57
Mean age (years)	65.2 ± 11.9	67.5 ± 9.0	0.32
(range)	(21–84)	39–90	
Gender (M/F)	31/26	26/32	0.31
Height (cm)	162.7 ± 8.9	164.9 ± 9.3	0.44
(range)	(137.6–183.0)	(141.0–178.7)	
Body weight (kg)	65.3 ± 11.7	67.1 ± 11.4	0.63
(range)	(38.5–105.7)	(42.0–82.3)	
BMI	23.9 ± 4.4	24.0 ± 3.9	0.90
(range)	(17.0–33.4)	(18.1–38.2)	
Diagnosis			0.58
Degenerative spondylolisthesis	31	33	
Spondylolytic spondylolisthesis	9	7	
Foraminal stenosis	17	18	
Level of fusion			0.73
L3–L4	2	1	
L4–L5	35	31	
L5–S1	20	26	

**Table 2 medicina-57-00150-t002:** Comparison of operative data between microendoscope-assisted (ME) and open groups.

	ME Group (*n* = 57)	Open Group (*n* = 58)	*p*
Operative time (min)	128.0 ± 34.9	176.6 ± 37.3	0.001
(range)	59–228	103–295	
Intraoperative blood loss (mL)	81.0 ± 92.8	315.2 ± 214.1	0.001
(range)	10–650	75–1080	
Number of cages	1.27 (1–2)	1.36 (1–2)	0.54
Cage type (PEEK/titanium)	55/2	54/4	0.41
Intraoperative complications	3	3	0.98
	all cases dural tear	all cases dural tear	
Duration of hospital stay (day)	10.7 ± 2.0	18.5 ± 6.1	0.001
(range)	(8–17)	(10–47)	
Postoperative complications	3	2	0.79
	cageloosening,	hematoma,	
	infection,	Vertebral body fracture	
	screw loosening		

PEEK: Poly Ether Ether Ketone.

**Table 3 medicina-57-00150-t003:** Comparison of patient-reported outcomes between microendoscope-assisted (ME) and open groups.

	ME Group	Open Group	*p*
NRS Low back pain			
Preoperative	5.89 ± 2.6	6.24 ± 3.0	0.76
Postoperative	2.42 ±1.9	3.51 ± 2.1	0.12
NRS Leg pain			
Preoperative	5.22 ± 3.2	5.77 ± 3.9	0.64
Postoperative	2.16 ± 1.8	2.76 ± 2.69	0.34
ODI			
Preoperative	20.8 ± 9.2	20.2 ± 10.8	0.89
Postoperative	6.9 ± 5.3	8.8 ± 5.8	0.44
EQ-5D			
Preoperative	0.56 ± 0.12	0.53 ± 0.15	0.71
Postoperative	0.79 ± 0.17	0.70 ± 0.18	0.51

Numerical Rating Scale (NRS), Oswestry disability index (ODI), EuroQol 5 Dimension (EQ-5D).

## Data Availability

The data presented in this study are available on request from the corresponding author.

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
