# Peer review of "Microendoscope-Assisted Versus Open Posterior Lumbar Interbody Fusion for Lumbar Degenerative Disease: A Multicenter Retrospective Cohort Study"

_medicina, 2021, doi:10.3390/medicina57020150_

Round 1

Reviewer 1 Report

I read your paper with interest, generally it is well written

In your abstract you state no studies have been done on MIS vs Open PLIF  are you sure about this. here are 2  examples that seem to discuss this:

PMID: 17334287, PMID: 19876659

In the abstract you note

3 patients in both groups have dural tears  but then you state that there are only 3 complications in one group and 2 in the other. A dural tear is a complication — therefore there should be at least 3 complications attributed to both groups.

methods -- I think it is important to detail a bit more the 2 surgical approaches and the differences in your techniques for the endoscopic and open approach.

Table 1 -- Should say L5-S1 not L5-S

Results/Discussion - is it possible that more fusion was obtained in the open group because fusion material, bone particles, Grafton was used to help with the fusion that was not used in the endoscopic group?

Author Response

(Q1) In your abstract you state no studies have been done on MIS vs Open PLIF  are you sure about this. here are 2  examples that seem to discuss this: PMID: 17334287, PMID: 19876659

 Thank you for your comment. As you pointed out, there were a few studies on the comparison between MIS and open PLIF. (line44-52) However, in these studies, the MIS procedure was performed using a microscope, while we are studying the technique using an endoscope. Therefore, we changed to emphasize that our study is about the microendoscope-assisted PLIF. (line23)

(Q2) In the abstract you note. 3 patients in both groups have dural tears  but then you state that there are only 3 complications in one group and 2 in the other. A dural tear is a complication — therefore there should be at least 3 complications attributed to both groups.

 Thank you for your advice. The dural tears is for intraoperative complications, and the descriptions in 3 and 2 are for postoperative complications. We have added these descriptions for clarity. (line30)

(Q3) I think it is important to detail a bit more the 2 surgical approaches and the differences in your techniques for the endoscopic and open approach.

Thank you for your comment. As you pointed out, we have added about surgical approaches. (line81-82)

(Q4) Table 1 -- Should say L5-S1 not L5-S

Thank you for your advice. As you pointed out, we have corrected that. (table1)

(Q5) Results/Discussion - is it possible that more fusion was obtained in the open group because fusion material, bone particles, Grafton was used to help with the fusion that was not used in the endoscopic group?

It was found that only autogenous bone or hydroxyapatite was used in our study and no Grafton was used. Since there is no use of Grafton in both two groups, we believe that there is no difference between the two groups due to the grafted material.

Reviewer 2 Report

The authors sought to compare ME assisted and OPLIF approaches of spine surgery for lumbar diseases. The study is well designed and the conclusions are well supported by the data. There are a few concerns that need to be addressed.

  1. There are several grammatical mistakes throughout the text that hinder reading, please do a through proofreading, citations should come before punctuations e.g. [3, 4]. Rather than after them e.g. .[3,4] This is an issue throughout the text.
  2. The standard deviation for these means need to be reported throughout, the range of variability for parameters such as the operation time are wide and without the standard deviation it is difficult to evaluate the robustness of the statistical analyses performed.
  3. As there is a considerable sample size in this study, why did the authors opt for a Mann-Whitney U? Was a test of homogeneity of variance conducted to warrant a non-parametric approach?
  4. Was there a statistical assessment of site-to-site/surgeon/surgeon variability? If microendoscopic mastery is key to a quick surgery this would likely result in significant variabilities between surgeons.
  5. If the functional recovery of the patients undergoing these two types of surgeries are similar, why are patients more satisfied in one compared to the other? Was there a survey that was completed that could shed more light on the subject?
  6. Currently, what decides which approach is used? Is there always an option between these two approaches? If patient satisfaction and risk of complications are similar, would ME still be considered a better approach just for the reduced operation time and blood loss?

Author Response

(Q1) There are several grammatical mistakes throughout the text that hinder reading, please do a through proofreading, citations should come before punctuations e.g. [3, 4]. Rather than after them e.g. .[3,4] This is an issue throughout the text.

Thank you for your advice. As you pointed out, we have changed all citations.

(Q2) The standard deviation for these means need to be reported throughout, the range of variability for parameters such as the operation time are wide and without the standard deviation it is difficult to evaluate the robustness of the statistical analyses performed.

 Thank you for your advice. We have added the standard deviation in the table 1.

(Q3) As there is a considerable sample size in this study, why did the authors opt for a Mann-Whitney U? Was a test of homogeneity of variance conducted to warrant a non-parametric approach?

We chose Mann-Whitney U because those were not normally distributed when examined statistically.

(Q4) Was there a statistical assessment of site-to-site/surgeon/surgeon variability? If microendoscopic mastery is key to a quick surgery this would likely result in significant variabilities between surgeons.

Thank you for your comment. In this study, we were not able to examine them statistically, because this study was a multicenter study involving six hospitals, which resulted in a large number of surgeons. Therefore, we added this to the limitation. (line241)

(Q5) If the functional recovery of the patients undergoing these two types of surgeries are similar, why are patients more satisfied in one compared to the other? Was there a survey that was completed that could shed more light on the subject?

Thank you for your comment. As shown on line 232, we speculated that less invasive procedures led to more satisfaction. However, it was difficult to speculate on the cause of this difference because this study was a multicenter study in which the procedures differed from institution to institution and this study’s questions including satisfaction were evaluated in numerical form.

(Q6) Currently, what decides which approach is used? Is there always an option between these two approaches? If patient satisfaction and risk of complications are similar, would ME still be considered a better approach just for the reduced operation time and blood loss?

Based on the results of our study, we believe that ME-PLIF is a better approach. However, since there are only a limited number of facilities that can perform the ME-PLIF at present, we choose to perform the ME-PLIF in all cases where we can.

Round 2

Reviewer 1 Report

Minor points:

- the sentence you added in the methods section is not entirely clear “Firstly a 20-mm incision was made into the skin 15mm symptomatic outside from the midline on the level of the intervertebral disc. Maybe you didn’t mean to put “symptomatic” in there?

- please create a subheading for your limitations paragraph in the discussion